# Nodding syndrome, a case-control study in Mahenge, Tanzania: *Onchocerca volvulus* and not *Mansonella perstans* as a risk factor

**Luís-Jorge Amaral**[1], **Dan Bhwana**[2], **Athanas D. Mhina**[2], **Bruno P. Mmbando**[2], **Robert Colebunders**[1]*

1 Global Health Institute, University of Antwerp, Antwerp Belgium, 2 National Institute of Medical Research, Tanga, Tanzania

* robert.colebunders@uantwerpen.be

## Abstract

### Background

Nodding syndrome (NS) has been consistently associated with onchocerciasis. Nevertheless, a positive association between NS and a *Mansonella perstans* infection was found in South Sudan. We aimed to determine whether the latter parasite could be a risk factor for NS in Mahenge.

### Methods

Cases of epilepsy were identified in villages affected by NS in Mahenge, Tanzania, and matched with controls without epilepsy of the same sex, age and village. We examined blood films of cases and controls to identify *M. perstans* infections. The participants were also asked for sociodemographic and epilepsy information, examined for palpable onchocercal nodules and onchocerciasis-related skin lesions and tested for anti-*Onchocerca volvulus* antibodies (Ov16 IgG4) by ELISA. Clinical characteristics of cases and controls, *O. volvulus* exposure status and relevant sociodemographic variables were assessed by a conditional logistic regression model for NS and epilepsy status matched for age, sex and village.

### Results

A total of 113 epilepsy cases and 132 controls were enrolled, of which, respectively, 56 (49.6%) and 64 (48.5%) were men. The median age in cases and controls was 28.0 (IQR: 22.0–35.0) and 27.0 (IQR: 21.0–33.3) years. Of the persons with epilepsy, 43 (38.1%) met the probable NS criteria and 106 (93.8%) had onchocerciasis-associated epilepsy (OAE). *M. perstans* infection was absent in all participants, while Ov16 seroprevalence was positively associated with probable NS (odds ratio (OR): 5.05, 95%CI: 1.79–14.27) and overall epilepsy (OR: 2.03, 95%CI: 1–07–3.86). Moreover, onchocerciasis-related skin manifestations were only found in the cases (n = 7, p = 0.0040), including persons with probable NS (n = 4, p = 0.0033). Residing longer in the village and having a family history of seizures

**Data Availability Statement:** All data underlying the findings described in this manuscript are freely

available to other researchers. See supplementary information (S1 dataset)

**Funding:** The study was funded by VLIR-UOS (Flemish University development cooperation) under grant number 671055 to RC, Research Foundation Flanders (FWO) under grant number G0A0522N to RC, and La Caixa Foundation under the grant number B005782 to LJA. The funders had no role in study design, data collection and analysis, decision to publish, or preparation of the manuscript.

**Competing interests:** The authors have declared that no competing interests exist.

were positively correlated with Ov16 status and made persons at higher odds for epilepsy, including probable NS.

## Conclusion

In contrast to *O. volvulus*, *M. perstans* is most likely not endemic to Mahenge and, therefore, cannot be a co-factor for NS in the area. Hence, this filaria is unlikely to be the primary and sole causal factor in the development of NS. The main risk factor for NS remains onchocerciasis.

### Author summary

Nodding syndrome (NS) is a rare and severe form of epilepsy that mainly affects children. The syndrome has only been found in onchocerciasis-endemic areas, and epidemiological studies suggest that it may be one of the forms of onchocerciasis-associated epilepsy caused by the parasite *Onchocerca volvulus*. Nevertheless, other causes have been considered, such as an infection with *Mansonella perstans*. To identify a main causal factor for NS, it needs to be present in all sites where the syndrome is found. Hence, our team explored 113 people with epilepsy and 132 healthy controls in Mahenge, Tanzania, to determine if *M. perstans* could be behind the epidemic of NS in the area. Over a third of the epilepsy cases in the study met the criteria for probable NS. None of the participants was diagnosed with an *M. perstans* infection. However, those exposed to onchocerciasis and with onchocerciasis-associated skin lesions were more likely to have epilepsy, including probable NS. In conclusion, onchocerciasis remains the most likely main risk factor for NS.

## Introduction

Nodding syndrome (NS) is a rare and severe form of epilepsy that mostly affects children in sub-Saharan Africa [1]. A link between NS and onchocerciasis has been suspected since the former was first described in the 1960s [2]. More recent epidemiological studies suggest that NS is one of the clinical manifestations of onchocerciasis-associated epilepsy (OAE) [3]. Onchocerciasis is a disease caused by the filarial nematode *Onchocerca volvulus*, spread by blackflies (*Simulium* spp.) and that leads to skin and eye disease [4]. Onchocerciasis-associated epilepsy appears in previously healthy children between the ages of 3 and 18 years, with a peak age of onset around 8–11 years [3]. The epidemiology of epilepsy in onchocerciasis-endemic areas differs from the non-endemic regions in Africa, where most epilepsy develops in children before the age of three years due to perinatal causes and genetic antecedents [5,6] or in adulthood due to brain tumours, strokes and neurocysticercosis [7,8]. OAE presents a broad spectrum of clinical manifestations, including NS and Nakalanga syndrome [9]. Despite a large number of epidemiological studies showing the association between onchocerciasis and epilepsy, there is a reluctance to consider this association causal, mainly because there is little evidence that *O. volvulus* microfilariae can penetrate the brain [10] and because, so far, no indirect mechanism has been identified to explain the pathogenesis of OAE [9]. Co-factors could also play a role in the pathogenesis of OAE. A study in South Sudan suggested that certain human leukocyte antigen (HLA) types could be a risk factor for developing NS, while

other HLA types could be protective [11]. It has also been suggested that parasitic co-infections could increase the risk of developing epilepsy [12].

Another filarial infection potentially associated with NS is *Mansonella perstans* [13]. In three case-control studies conducted between 2001 and 2002 in Mundri, Western Equatoria State in South Sudan, not only a positive association was found between NS and an *O. volvulus* infection (odds ratio (OR): 9.2, p-value (p) <0.001), but also between NS and *M. perstans* infection (OR: 3.2, p = 0.005) [13]. However, in a case-control study conducted in 2014 in Titule, an onchocerciasis-endemic area in Bas-Uéle in the Democratic Republic of Congo, a *M. perstans* infection was not more prevalent in persons with epilepsy than in healthy controls (p = 0.91). Moreover, cases were significantly more burdened with onchocerciasis-related skin disease (p<0.01) [14].

Despite being the least studied, *M. perstans* is considered the most prevalent human filaria [15]. The infection is spread by midges (*Culicoides* spp.) and is believed to be prevalent in Western, Eastern and Central Africa and neotropical regions in South America [15]. The spectrum of clinical symptoms and signs remains ill-defined for *Mansonella* infections. Various clinical manifestations have been reported, including itching, swelling, joint pain, enlarged lymph nodes, vague abdominal symptoms and eosinophilia, but mansonellosis is often asymptomatic [15]. Moreover, it is frequently difficult to attribute clinical manifestations to a *M. perstans* infection due to the high frequency of co-infections with other parasites [15,16].

The diagnosis of *M. perstans* relies on detecting blood circulating microfilariae, real-time PCR or loop-mediated isothermal amplification assays (LAMP) testing [17]. Antibody-detecting enzyme-linked immunosorbent assay tests are available but not specific for *Mansonella* spp. [17].

During the International Scientific Meeting on NS organised in 2012 by WHO in Kampala, it was recommended that further investigation was needed to study *M. perstans* and *O. volvulus* infections as potential risk factors for NS [18]. Therefore, we investigated whether an infection with *M. perstans* could be a risk factor in the development of NS in villages affected by NS in Mahenge, Tanzania. We also investigated the exposure to *O. volvulus* infection.

## Methods

### Ethics statement

The study was conducted according to the guidelines of the Declaration of Helsinki. Ethical approval was obtained from the Ethics Committee of the National Institute for Medical Research, Tanzania (NIMR/HQ/R.8a/Vol.IX/3746) and the Ethics Committee of the Antwerp University Hospital, Belgium (B300201837863).

### Study settings

Between September 2021 and March 2022, we conducted a matched case-control study in Mahenge, Ulanga District, Tanzania. Persons with epilepsy identified in previous door-to-door surveys in 2017, 2018 and 2021 were selected from a suburban village (Vigoi) and nine rural villages in the Mahenge area known to have a high prevalence of epilepsy (Mzelezi, Mdindo, Sali and Msogezi, included in previous epilepsy studies [19,20], and Ebuyu, Euga, Isyaga, Isongo and Mgolo villages) (Fig 1).

The Mahenge Mountains are of high altitude, abundant in banana trees and rich in fast-flowing rivers, the latter providing suitable breeding sites for blackflies (locally known as *vifuna* in *Kiswahili*) [22]. The area was one of the most onchocerciasis-endemic foci in Tanzania before the implementation in 1997 of annual community-directed treatment with ivermectin (CDTI) [23,24,25]. The latter is the main intervention to control and eliminate

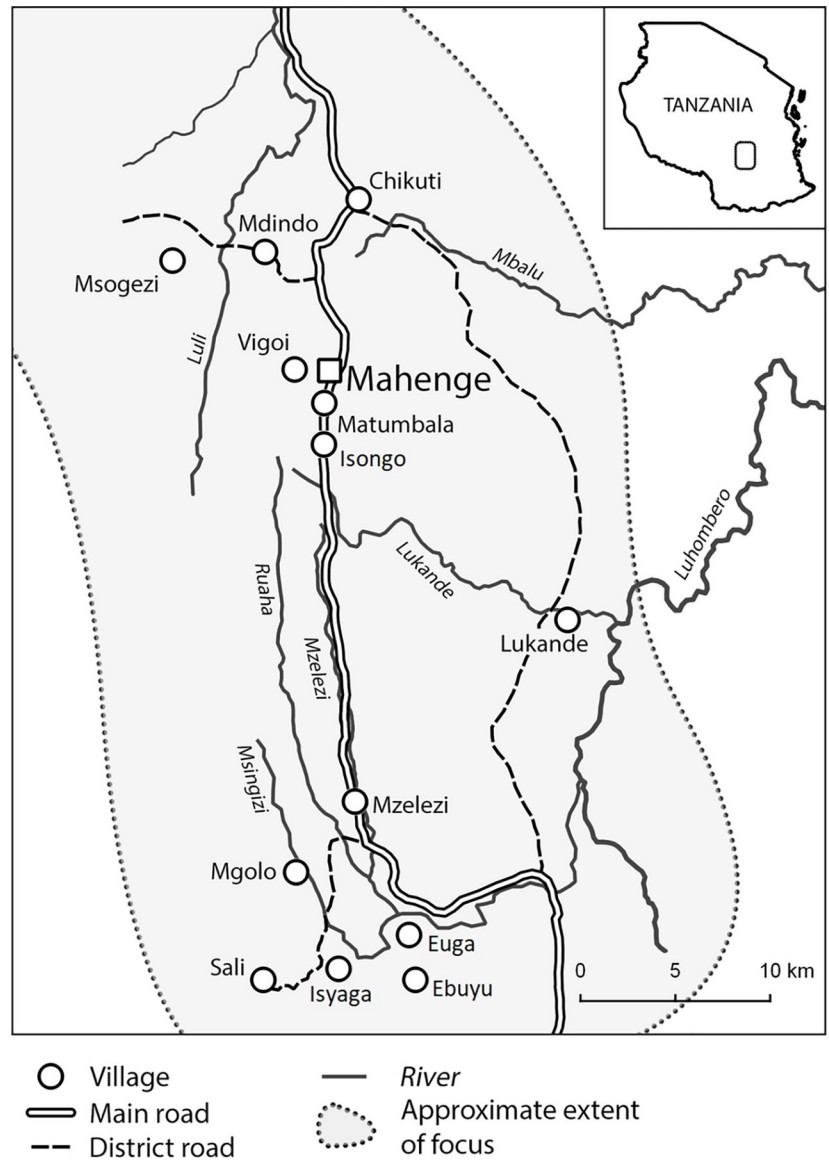

**Fig 1. Map of the Mahenge area and surveyed villages [21].**

onchocerciasis and, in 2019, was strengthened from annual to biannual in Mahenge [19]. Most of the local population belongs to the Wapogoro ethnicity and subsists on agriculture, livestock (chickens, goats and pigs, the latter mostly kept indoors and in suburban villages) and gem mining. *Loa loa* and *Wuchereria bancrofti* filarial infections are not believed to be prevalent in the area [26,27].

## Study design

The study was designed to be a case-control study where cases and controls were matched for age, sex and village. A case of epilepsy was defined according to the International League Against Epilepsy as a person with a history of at least two unprovoked seizures 24 hours or more apart [28]. Epilepsy cases and controls without epilepsy were people who screened positive or negative for epilepsy in the previous door-to-door surveys, respectively [29]. Persons with

epilepsy, including NS and Nakalanga syndrome, were individually matched with controls by age (± three years), sex and village (if possible, from the same neighbourhood). When a matching control was not available in the village of a case, one was sought in adjacent villages.

A case of probable NS was defined according to the 2013 modified consensus case definition of NS [30]. In brief, as a previously healthy three to 18 years old child who developed repetitive episodes of involuntary drops of the head to the chest on two or more occasions a day or more apart. They also had to present at least one of the following criteria: 1) neurological abnormalities; 2) clustering in space and time with other cases; 3) stunting or wasting; 4) delayed sexual or physical development or; 5) psychiatric symptoms. Overall, a probable case of NS was a person that met the NS criteria but in which the nodding episode was not documented by a trained healthcare worker, videotaped or shown by EEG/EMG.

A case of Nakalanga syndrome was defined as a person with a combination of stunted growth, delayed or absent external signs of secondary sexual development (endocrine dysfunction) and mental impairment, often associated with epileptic seizures and facial and thoracic dysmorphia [31].

A person with epilepsy was recognised with OAE if the following criteria were met: 1) epilepsy onset between the ages of three and 18 years; 2) without an obvious cause of epilepsy, and; 3) living for at least three years in the study area, known to be endemic for onchocerciasis and to have geographic and familial clustering of epilepsy cases [19,32].

Participants were asked to identify their closest family member with epilepsy, if any. Family history of epilepsy was then organised in three categories based on blood relationship (consanguinity): 1) First-degree relative with epilepsy (parent, sibling or son/daughter, with approximately 50% of their genes shared with the case/control); 2) Second-degree relative with epilepsy (uncle, cousin, grandparent or nephew, with approximately 25% of their genes shared with the case/control); and 3) in-law with epilepsy (brother/sister-in-law, mother/father-in-law, daughter/son-in-law, with no genetic link with the case/control).

## Sample size

In Mundri County, South Sudan, a significant association was found between NS and an *M. perstans* infection, with an odds ratio of 3.2 (p = 0.005). Among the NS cases, 52% (36/69) were infected with *M. perstans*, compared to 31% (20/65) controls without epilepsy [13]. Assuming a similar prevalence in Mahenge, at least half (50%) of NS cases and one-fourth (25%) of controls are expected to have *M. perstans* infection. Hence, at least 44 cases and 132 controls should be enrolled in the study, with a 1:3 matching ratio.

## Study procedures

After obtaining written informed consent, participants or their guardians were interviewed in the Swahili (*Kiswahili*) language. They were questioned about their sociodemographic and clinical characteristics, such as ethnicity, epilepsy (*kifafa* in *Kiswahili*) status, head nodding history (*amesinzia kichwa* in *Kiswahili*) and ivermectin intake. The participants then underwent a physical examination by a medical doctor (DB). The latter included detecting palpable onchocercal nodules (onchocercomata) and dermatological lesions. The mothers of the participants present were also interviewed and examined for the presence of palpable onchocercal nodules and dermatological lesions after signing informed consent.

A finger prick of blood was obtained from all cases and controls to make a dried blood spot and thick and thin films. Samples were collected, air-dried and stored at 4˚C until handled for further analysis to identify *M. perstans* infection and *O. volvulus* exposure. The latter involved the detection of IgG4 antibodies recognising *O. volvulus* Ov16 antigen in a dried blood spot by

enzyme-linked immunosorbent assay (ELISA), using the commercially available Ov16 IgG4 ELISA kits (Standard Diagnostics Gyeonggi-do, Republic of Korea) and following its manufacturer instructions. In turn, the procedure for detecting *M. perstans* infection encompassed staining the dried films with Giemsa and examining them for microfilariae under a microscope with a high-power objective lens. The sensitivity of a microscopic examination of a blood smear to detect a *M.* perstans infection is estimated to be 78% [33], while the specificity is close to 100% when a well-trained laboratory technician performs the test.

Skin snips were not obtained for *O. volvulus* microfilariae detection, as nearly all study participants took ivermectin during annual CDTI. The latter has been shown to mask the association between onchocerciasis and epilepsy [34].

### Statistical analysis

The sociodemographic and clinical characteristics of cases and controls were described with median and interquartile range (IQR) for continuous variables and number and frequency for categorical data. Differences in sociodemographic variables between cases and controls were studied using the chi-squared test for categorical variables (or Fisher's exact test if counts <5) and the t-test for continuous variables. Variables were also checked for correlation.

A conditional logistic regression model incorporating the case-control matching factors (sex, age and village) for epilepsy status was run for each clinical characteristic of cases and controls, *M. perstans* infection, *O. volvulus* exposure and relevant sociodemographic variables. Odds ratios (ORs), 95% confidence intervals (95% CIs) and p-values were presented. Some epilepsy cases were matched with controls from neighbouring villages, as no controls were available within the same village. Hence, a sensitivity analysis of the village matching was performed by adjusting the regression for the village of cases and controls. Variables with < 5% frequencies were compared using Fisher's exact test instead of the regression and p-values presented. Correlations between epilepsy predictors were explored, and a logistic regression adjusted for age, sex and village type was used to explore the association between Ov16 seropositivity and a family history of seizures.

The conditional model and Fisher's exact test were implemented a second and a third time for probable NS and for OAE by excluding from the analysis the persons with epilepsy that did not meet the criteria of these definitions. The controls of the persons with epilepsy not meeting the NS and OAE criteria were individually matched to the closest persons with epilepsy meeting the former, respectively. Two-tailed p-values below 0.05 were considered statistically significant. Analyses were performed using R and RStudio (4.2.1 and 2022.06.23 versions, respectively).

## Results

### Sociodemographic and clinical characteristics of the study population

The median age of the study population was 28.3 (IQR: 22.0–34.0) years, with 125 (51%) females and 120 (49%) males (Table 1). Cases of epilepsy and controls without the condition were well-matched for sex (p = 0.97), age (p = 0.64), village type (1.00) and ethnicity (p = 0.86). Nevertheless, no controls were available to be matched by age and sex for 22 cases within certain villages, particularly Mzelezi [10] and Isyaga [5] villages. Hence, these cases were matched with controls from neighbouring villages (S1 Table). The search for persons without epilepsy led to the recruitment of slightly more controls (132) than cases (113), with 19 epilepsy cases (16.8%) matched to two controls.

Nearly all participants reported taking ivermectin in the past. Families of persons with epilepsy have lived more years in the villages under study (p<0.0001). Persons with epilepsy were also

**Table 1. Sociodemographic characteristics of epilepsy cases and controls.**

| Variable | | Cases n = 113 | Controls n = 132 | p-value |
|---|---|---|---|---|
| **Male**, n (%) | | 56 (49.6) | 64 (48.5) | 0.97 |
| **Age**, median (IQR) | | 28.0 (22.0–35.0) | 27.0 (21.0–33.3) | 0.64 |
| **Village type**, n (%) | | | | 1.00 |
| Rural[a] | | 106 (93.8) | 124 (93.9) | |
| Sub-urban[b] | | 7 (6.2) | 8 (6.1) | |
| **Ethnicity**, n (%) | | | | 0.86 |
| Wapogoro | | 110 (97.3) | 130 (98.5) | |
| Other ethnicities [c] | | 3 (2.7) | 2 (1.5) | |
| **Ivermectin intake**, n (%) | | 105 (92.9) | 126 (95.5) | 0.57 |
| **Closest family member with epilepsy**, n (%) | | | | <0.0001 |
| First-degree relative with epilepsy | | 17 (15.0) | 13 (9.9) | 0.30 |
| | Parent | 3 | 0 | 0.097 [e] |
| | Sibling | 14 | 11 | 0.40 |
| | Son | 0 | 2 | 0.50 [e] |
| Second-degree relative with epilepsy | | 24 (21.2) | 4 (3.0) | <0.0001 |
| | Grandparent | 3 | 0 | 0.097 [e] |
| | Uncle/Aunt | 13 | 2 | 0.0029 |
| | Cousin | 4 | 1 | 0.19 [e] |
| | Nephew | 4 | 1 | 0.19[e] |
| In-law with epilepsy[d] | | 8 (7.1) | 3 (2.3) | 0.13 |
| No family members with epilepsy | | 64 (56.7) | 112 (84.9) | <0.0001 |
| **Consanguine marriage within the family, n (%)** | | 9 (8.0) | 5 (3.8) | 0.26 |
| **Years family lived in the village**, median (IQR) | | 28.0 (22.0–35.0) | 25.0 (20.0–31.3) | <0.0001 |

n—Number; %—Percentage frequency; IQR—Interquartile range.

[a] Villages Ebuyu, Euga, Isongo, Isyaga, Mdindo, Mgolo, Msogezi, Mzelezi and Sali.

[b] Village Vigoi.

[c] Other ethnicities identified were Hehe and Mndamba.

[d] A brother-/sister-in-law, mother-/father-in-law, daughter-in-law.

[e] Fisher's exact test was used to account for the rare frequency.

more likely to have a family member with epilepsy (p<0.0001). Most of the first-degree relatives with epilepsy were siblings (25, 83.3%), followed by mothers (2, 6.7%), sons (2, 6.7%) and father (1, 3.3%). Second-degree relatives with epilepsy were uncles (15, 53.5%), cousins (5, 17.9%), nephews (5, 17.9%) and grandparents (3, 10.7%). Over half of the in-laws with epilepsy were brothers-in-law (7, 63.6%), followed by daughter- (1), father- (1), mother- (1) and sister-in-law (1).

## Clinical characteristics of persons with epilepsy

Forty-three (38.1%) epilepsy cases met the criteria of probable NS, and 106 (93.8%) had OAE (Table 2). Those without OAE either developed epilepsy in adulthood (4), were new to the study area (1), had an obvious cause of epilepsy (1) or did not remember when they developed their first seizure (1).

The median age of seizure onset was ten years (IQR: 8.0–13.0), and most persons with epilepsy (74.3%) were under antiseizure treatment, usually phenobarbital (67, 79.8%). The median age of seizure onset was higher for other forms of epilepsy (11.0, IQR: 9.0–14.0) than probable NS (9.0, IQR: 7.5–11.0) (p = 0.0004).

**Table 2. Epilepsy and treatment characteristics of persons with epilepsy.**

| Variable | Epilepsy cases n = 113 |
|---|---|
| **Onchocerciasis-associated epilepsy,** n (%) | 106 (93.8%) |
| **Probably Nodding syndrome,** n (%) | 43 (38.1%) |
| **Nakalanga syndrome**, n (%) | 5 (4.4%) |
| **Age of seizure onset,** median (IQR) | 10.0 (8.0–13.0) |
| **Antiseizure treatment,** n (%) | 84 (74.3%) |

n—Number; %—Percentage frequency; IQR—Interquartile range.

### *Mansonella perstans* infection and *Onchocerca volvulus* exposure status in epilepsy cases and controls

None of the 113 epilepsy cases (including 43 with probable NS) and 132 controls were diagnosed with an *M. perstans* infection. (Table 3). Ov16 seroprevalence was associated with epilepsy (OR: 2.03, 95% CI: 1.07–3.86), including OAE (OR: 2.26, 95% CI: 1.15–4.47) and probable NS (OR: 5.05, 95% CI: 1.79–14.27). Onchocercal skin manifestations were only found in persons with epilepsy (6.2%, p = 0.0040), especially among probable NS cases (9.3%, p = 0.0033). Cases and controls had a lower prevalence of onchocercal nodules than their mothers (p = 0.018). Participants with second-degree relatives with epilepsy were at higher odds of having epilepsy, including the ones with OAE and probable NS. Moreover, persons with OAE, including probable NS, were more likely to have an in-law with epilepsy. Living longer in the village was also a risk factor for epilepsy (p = 0.016), including OAE (p = 0.0024) and probable NS (p = 0.0023). The sensitivity analysis performed for the village matching did not impact the significance of any findings.

The length of residence in the village was correlated with Ov16 seroprevalence (cor = 0.30, p<0.0001), and the latter was also correlated to a family history of seizures (cor = 0.32, p<0.0001). When added to a univariable logistic regression adjusted for age, sex and village type, having a first-degree relative with epilepsy was strongly associated with Ov16 seropositivity (adjusted OR: 15.02, 95% CI: 2.97–274.7; p = 0.0094) but not significantly related to having a second-degree relative (p = 0.21) or an in-law with epilepsy (p = 0.25). Similarly, a family history of seizures lost all significance when an interaction effect with Ov16 positivity was added to the conditional logistic regression for epilepsy, OAE or NS.

Epilepsy cases were more likely to have a relative of a previous generation with epilepsy than controls (grandparent, parent or uncle/aunt, 16.8% versus 1.5%, p<0.0001). Moreover, most epilepsy cases (92.3%, 12/13) and all controls (100.0%, 10/10) with a sibling with epilepsy were Ov16 seropositive (one case and one control were not tested).

## Discussion

To identify the cause of NS, a common risk factor must be found in all areas where NS has been reported. So far, the only risk factor present in all those areas is onchocerciasis. We explored the possible association between *M. perstans* and NS in Mahenge, where NS and Nakalanga syndrome have been known to occur since the 1960s. We also aimed to confirm the association between onchocerciasis and NS.

None of the 132 controls and 113 persons with epilepsy, including 43 with probable NS, were diagnosed with an *M. perstans* infection. Therefore, *M. perstans* is likely not endemic in Mahenge and should no longer be considered a risk factor for NS in the area. In contrast,

**Table 3. Relevant sociodemographic variables, *M. perstans* infection and *O. volvulus* seroprevalence, ochocercal nodules and skin manifestations among epilepsy cases, including OAE and probable NS, and controls.**

| Variable | Epilepsy cases (n = 113) | OAE cases (n = 106) | pNS cases (n = 43) | Controls (n = 132) | Level of association (OR) | | | |
|---|---|---|---|---|---|---|---|---|
| | | | | | Epilepsy cases (95% CI) | OAE cases (95% CI) | pNS cases (95% CI) | p-value (Epilepsy / OAE / pNS) |
| ***M. perstans* blood smear** (positive) | 0.0% (0 out 113) | 0.0% (0 out 106) | 0.0% (0 out 43) | 0.0% (0 out of 132) | _[b] | _[b] | _[b] | _[b] |
| **Ov16 ELISA** (positive) | 75.0% (78 out of 104) | 76.3% (74 out of 97) | 84.6% (33 out of 39) | 60.0% (72 out of 120) | 2.03 (1.07–3.86) | 2.26 (1.15–4.47) | 5.05 (1.79–14.27) | 0.030 / 0.019 / 0.0023[a] |
| **Onchocercal nodules** (one or more) | 17.7% (20 out of 113) | 17.9% (19 out of 106) | 16.3% (7 out of 43) | 11.4% (15 out of 132) | 1.80 (0.82–3.97) | 2.02 (0.89–4.60) | 1.90 (0.56–6.46) | 0.15 / 0.093 / 0.31[a] |
| **Onchocercal skin manifestations**[f] (one or more) | 6.2% (7 out of 113) | 4.7% (5 out of 106) | 9.3% (4 out of 43) | 0.0% (0 out of 132) | - | - | - | 0.0040 / 0.017 / 0.0033[g] |
| **Onchocercal nodules of mothers** (one or more) | 42.9% (30 out of 70) | 43.9% (29 out of 66) | 38.2% (13 out of 34) | 38.8% (19 out of 49) | 2.53 (0.97–6.60) | 2.43 (0.92–6.39) | 1.46 (0.49–4.40) | 0.059 / 0.073 / 0.50[a] |
| **Onchocercal skin manifestations of mothers**[f] (one or more) | 11.4% (8 out of 70) | 10.6% (7 out of 66) | 11.8% (4 out of 34) | 12.3% (6 out of 49) | - | - | - | 1.00 / 0.78 / 1.00[g] |
| **Closest family member with epilepsy** | | | | | | | | |
| First-degree relative with epilepsy[c] | 17.7% (20 out of 113) | 17.9% (19 out of 106) | 18.6% (8 out of 43) | 9.9% (13 out of 132) | 1.76 (0.81–3.81) | 2.16 (0.95–4.93) | 1.63 (0.52–5.12) | 0.15 / 0.067 / 0.40[a] |
| Second-degree relative with epilepsy[d] | 18.9% (21 out of 113) | 18.9% (20 out of 106) | 14.0% (6 out of 43) | 3.0% (4 out of 132) | 10.40 (3.04–35.60) | 10.77 (3.12–37.25) | 11.78 (2.84–48.89) | 0.0002 / 0.0002 / 0.0007[a] |
| In-law with epilepsy[e] | 7.1% (8 out of 113) | 7.5% (8 out of 113) | 7.0% (3 out of 43) | 2.3% (3 out of 132) | 4.14 (1.00–17.18) | 4.35 (1.04–18.20) | 11.04 (1.39–87.43) | 0.051 / 0.040 / 0.023[a] |
| **Years family lived in the village**, median (IQR) | 28.0 (22.0–35.0) | 26.5 (22.0–33.0) | 28.0 (20.5–31.5) | 25.0 (20.0–31.3) | 1.05 (1.01–1.09) | 1.05 (1.01–1.09) | 1.07 (1.01–1.13) | 0.016 / 0.024 / 0.023[a] |

n—Number; OAE–Onchocerciasis-associated epilepsy; pNS–Probable nodding syndrome; OR–Odds ratio; 95% CI– 95% Confidence interval.

[a] Conditional logistic regression model matched for village, age and sex.

[b] No statistical test was used, as all controls and cases tested negative for *M. perstans*

[c] A parent, sibling or son.

[d] A uncle, nephew, grandparent or cousin.

[e] A brother-/sister-in-law, mother-/father-in-law, daughter-in-law.

[f] Leopard skin; dry, thickened, wrinkled skin; papular rash.

[g] Fisher's exact test was used to account for the rare frequency.

Ov16 seroprevalence (OR: 2.03, 95% CI: 1.07–3.86) and onchocercal skin manifestations (p = 0.0040) were positively associated with cases of epilepsy, including probable NS (OR: 5.05, 95% CI: 1.79–14.27, and p = 0.0033, respectively) and OAE (OR: 2.26, 95% CI: 1.15–4.47, and p = 0.017, respectively). Moreover, families of persons with Ov16 antibody seropositivity lived for longer in the study villages (p<0.0001), which are known to have been meso- to hyperendemic for onchocerciasis before CDTI and still have persistent transmission of *O. volvulus* [19,20].

A person with a second-degree relative with epilepsy had higher odds of having epilepsy, including probable NS. This observation is consistent with the fact that family members,

especially first-degree relatives, often live in the same neighbourhood and thus share similar exposures to *O. volvulus*-infected blackflies [35]. This is supported by the fact that individuals who tested Ov16 seropositive were more likely to have a first-degree relative with epilepsy. Notably, almost all epilepsy cases (92.2%) and all controls with a sibling with epilepsy tested seropositive. This finding further suggests that the association between epilepsy and direct family members with the condition is likely related to onchocerciasis exposure. The lack of a significant association between epilepsy and having a first-degree relative with the condition can be attributed to the comparable exposure of cases and controls to blackfly bites and the small sample size of the study. In contrast, a significant association was observed between being a second-degree relative and epilepsy (p<0.001). The increased presence of uncles/aunts, cousins and nephews with epilepsy in case families can likely be attributed to their shared living environment or proximity to the cases, placing them at similar risk for developing OAE.

There were at least two adults with epilepsy in 14 (10.8%) of the 113 case families, while in 19 (9.9%) of 132 control families, there was at least one adult with epilepsy. The latter suggests that control families were likely exposed to a lesser extent and duration to infected blackfly bites than the families of cases, as evidenced by the higher Ov16 seroprevalence of cases and longer family residence in the study area (p<0.05). Hence, it is anticipated that cases would exhibit higher microfilarial loads [36].

A person with an in-law with epilepsy had higher odds of having OAE (OR: 4.35, 95% CI: 1.04–18.20) or NS (OR: 11.04, 95% CI: 1.39–87.43). The latter may also be explained by the in-law residing in the same high-risk area for onchocerciasis as the cases and the impact of epilepsy-related stigma. The stigma surrounding epilepsy can lead to social challenges, including finding a marital partner for individuals with epilepsy and their family members [37,38,39]. For instance, a recent study in Mahenge reported that a mere 15% of persons with epilepsy were married [40]. It is, therefore, plausible to consider that individuals from families affected by epilepsy may be willing to marry a person with epilepsy. Furthermore, it has been shown that persons with epilepsy in Africa frequently marry community members of lower social status, further emphasising the influence of social dynamics in forming relationships [39].

The higher epilepsy prevalence among relatives of cases may indicate that the clustering of epilepsy in families could also be due to a potential genetic predisposition, where certain families could be more susceptible to both *Onchocerca* parasite infection and NS development. Still, this association between epilepsy and having a family member with epilepsy was lost when the conditional regression was adjusted for Ov16 seropositivity, indicating that exposure to *O. volvulus* played a more significant role in the observed relationship.

Only 17.9% of the OAE cases, 16.3% of the probable NS cases and 11.4% of controls had palpable onchocercal nodules. Given the more than 20 years of distributing ivermectin to control onchocerciasis, this low prevalence of onchocercal nodules was expected. Indeed, fewer nodules were observed among the cases and controls than among their mothers. In onchocerciasis-endemic areas, the number of nodules increases with age and, therefore, may have been easier to palpate in the mothers [41]. The lack of association between the onchocercal nodules prevalence of participants and epilepsy could also be related to the low sample size and the difficulty of palpation of a low number of nodules. Ultimately, nodule palpitation is a less reliable methodology in settings with onchocerciasis control and younger populations, as they usually have fewer nodules to be detected [41]. A study in an onchocerciasis hyperendemic focus in Cameroon before the implementation of CDTI found a positive association between onchocercal nodules and cases of epilepsy [42]. In contrast, similar to our findings, a study in an onchocerciasis hyperendemic area in northern Uganda under CDTI found a lack of association between NS and onchocercal nodules but a strong association with a positive onchocerciasis serology [43].

Onchocercal skin manifestations were only prevalent in persons with epilepsy, including with probable NS. However, these were mainly chronic lesions because of a past *O. volvulus* infection (e.g. leopard skin; dry, thickened, wrinkled skin). The median age of the cases of epilepsy was 28.0 years (IQR: 22.0–35.0), while the median age of onset of epilepsy was 10.0 (IQR: 8.0–13.0), the latter as expected for OAE. This age difference of 18 years suggests that the incidence of OAE in Mahenge has recently decreased.

Regarding *M. perstans*, there is no recent information on its epidemiology in Tanzania. Previous research has identified the parasite between Lakes Tanganyika and Lake Victoria, particularly in areas with banana plantations and abundant rainfall, as the *Culicoides* vector can grow on decaying banana trees [44]. To our knowledge, no *M. perstans* studies have been conducted in Mahenge. The presence of the *Culicoides* vector in the area can be inferred from a report on the Schmallenberg virus in sheep and goats, also spread by this vector [45]. Our study shows the absence of *M. perstans* in Mahenge, which may be due to the unique ecology of the area, with a lower temperature than mainland Tanzania, higher altitude and perhaps less competent or anthropophilic *Culicoides* spp. It is very unlikely that *M. perstans* was in Mahenge before, as there was no control against the parasite. Ivermectin has little to no impact on the parasite [46,47] and does not reduce its prevalence [48]. Diethylcarbamazine and doxycycline were never distributed in Mahenge. As lymphatic filariasis is not endemic in the area, mebendazole is not given to adults; nevertheless, it is possible that preschool- to school-aged children receive the former drug or albendazole once or twice a year as part of the deworming programme [49,50]. Aside from the fact that our study population were adults (median: 28.3, IQR: 22.0–34.0), this treatment is unlikely to have had a substantial impact on *M. perstans* prevalence. Albendazole has been shown to exert little to no effect on a *M. perstans* infection, even in combination with ivermectin [17], while mebendazole should be given daily for several weeks to exert an effect [17,48].

Given the geographic distribution of *M. perstans* in sub-Saharan Africa, a main causal role with NS is very unlikely. Indeed, *M. perstans* infections are widespread in sub-Saharan Africa and are present in many areas where NS has never been reported. Furthermore, the distribution of *M. perstans* infections cannot explain the localisation of NS in villages and households close to rapid-flowing rivers and *O. volvulus* breeding sites [51,52]. In northern Uganda, in the area where a NS epidemic appeared around the year 2000 [53], the prevalence of *M. perstans* infection was very low and much lower than in other parts of Uganda where NS has never been reported [54]. Moreover, in a cohort study in an onchocerciasis-endemic area in Cameroon, the relative risks of children developing epilepsy depended on the *O. volvulus* microfilarial density in a dose-response way, but no association was found between the presence of *L. loa* or *M. perstans* microfilaremia and the development of epilepsy [55]. Nodding and Nakalanga syndromes were previously reported in an onchocerciasis and *M. perstans* co-endemic area in western Uganda. However, while *M. perstans* is still endemic in the area, NS and Nakalanga syndrome ceased to appear when *O. volvulus* transmission was eliminated by CDTI and vector control [13].

How to explain the association between an *M. perstans* infection and NS in South Sudan? One possibility could be that children with NS in South Sudan nearly never sleep under insecticide-impregnated bed nets together with other children because the local community believes that NS is transmissible through contact [56]. Therefore, children with NS are often isolated from the rest of the family and not allowed to eat from the same plate with other children or to sleep with them in the same room. The midges that transmit *M. perstans* are most active at night [57]. Midges are very small and hence are considered to pass through bed nets [58]. However, insecticide-treated nets have been shown to protect horses from *Culicoides*, the *M. perstans* vector, in the event of an African horse sickness virus epidemic [59]. Nevertheless, as

impregnated bed nets were not distributed in 2001–2002 in Mundri, bed nets are unlikely to explain the observed association between NS and *M. perstans* infection in the study by Tumwine *et al.* [13].

Factors associated with poverty also potentially played a role in the association between *M. perstans* infection and NS. Indeed, a study on Bioko Island, Equatorial Guinea, showed that people from the lowest socioeconomic quintile were five times more likely to be infected with *M. perstans* [60]. Children with NS and their families also belong to the poorest and most vulnerable groups in their communities [38].

The vector *Culicoides* presence is often associated with aquatic environments, banana plantations and plantain stems [57]. Families with children with NS generally live close to rapid-flowing rivers [61] and blackfly breeding sites, the vector of *O. volvulus* [62]. However, these sites are not known to be particular sites where *Culicoides* bite, possibly explaining why *M. perstans* infection is absent from many areas with NS prevalence.

Farming could be another explanation for the association between an *M. perstans* infection and NS. In Maridi, South Sudan, belonging to a farming family was found to be a risk factor for NS [29]. In a study in Nigeria, the prevalence of mansonellosis was significantly higher among rural dwellers (34.6%) than among urban dwellers (22.5%), and among farmers (59.8%) than in civil servants (7.6%) (p<0.05) [63]. Moreover, farming is also a risk factor for onchocerciasis, as many fertile lands are close to blackfly breeding sites and have high biting rates [29].

In a case-control study in northern Uganda, preterm birth was found to be a risk factor for NS [64]. As preterm birth could be related to an *O. volvulus* infection of the pregnant mother [65], it was suggested that an *O. volvulus* infection of the mothers could cause a "parasitic tolerance" in their offspring. This tolerance could induce a higher load of *O. volvulus* microfilarial infection after the children are exposed to *O. volvulus*-infected blackflies than if the mothers were free of infection during pregnancy [66]. This high-level *O. volvulus* infection in young children was documented as a risk factor in onchocerciasis endemic areas in Cameroon for developing epilepsy [55,67]. It should be investigated whether an *O. volvulus* infection of mothers can also induce "parasitic tolerance" to an *M. perstans* infection. For instance, concomitant infection with *O. volvulus* and *M. perstans* was common in rural villages of southern Cameroon, while the opposite was seen with *Loa loa* [68], the latter possibly due to distinct vector ecologies [69]. Another study in central Cameroon found a positive association between *O. volvulus* and *L. loa* infections [69]. Hence, concomitant filarial infections may increase the tolerance of the human host to these parasites. Further studies could be conducted to explore this hypothesis among the different filariae.

We also need to mention the limitations of our study. Firstly, while controls were chosen among volunteers without epilepsy of the same age and area as the cases, the selection was not entirely at random. Due to the COVID-19 complot theories circulating in the community at the time of the study, it was difficult to find village-matched controls for every case in certain villages. We, therefore, added controls from neighbouring villages to achieve the desired statistical power. We mitigated this by performing a sensitivity analysis for the village of the participants in the regression to ensure a robust matching of the study population. Secondly, nearly all participants have taken ivermectin (94.3%). Therefore, no strong association between onchocerciasis and epilepsy could be documented. Thirdly, the association found between anti-*O. volvulus* antibodies and NS cases may have been stronger if a more sensitive test, such as the *O. volvulus* luciferase immunoprecipitation system (LIPS) assay, had been used instead of the Ov16 ELISA. In northern Uganda, ELISA and LIPS detected, respectively, 66.7% and 94.9% *O. volvulus* seropositivity in NS cases and 31.8% and 48.8% in healthy controls [43].

A fourth limitation pertains to the diagnostic test used to assess *M. perstans* infection status. While a test with higher sensitivity, such as PCR [33], could have been employed, all cases and

controls tested negative for *M. perstans* infection using the blood smear, which has an esti-
mated sensitivity of 78% [33]. This suggests the parasite is either absent from the region or
present at very low levels. Lastly, it would have been beneficial to gather more detailed infor-
mation regarding the family history of epilepsy, including the number of family members
affected and their ages. Despite these limitations, our study provides valuable insights into the
relationship between onchocerciasis, *M. perstans*, and NS. Further research addressing these
limitations and exploring other potential risk factors is warranted to enhance our understand-
ing of the pathogenesis of nodding syndrome and improve interventions for affected
populations.

## Conclusion

The first step to understanding the pathogenesis of NS is to identify a common risk factor
across all sites where this condition is reported. Our study in Mahenge indicates that *M. per-
stans* is unlikely to be endemic in Mahenge and, therefore, cannot play a co-factor role in
developing NS in the area. While our findings suggest that *M. perstans* is unlikely to be the
main and sole cause of NS, it is important to acknowledge that *M. perstans* could still poten-
tially be a co-risk factor for NS in areas where the parasite is endemic. Evidently, more research
on the parasite is needed to determine its ecology, pathogenicity, and link with other infec-
tions, including onchocerciasis.

Onchocerciasis remains the main risk factor for NS and other forms of epilepsy that meet
the criteria for OAE. However, the specific mechanisms by which the *O. volvulus* parasite may
lead to epilepsy remain to be identified. Potential explanations include the occasional passage
of *O. volvulus* microfilariae across the blood-brain barrier in children with a high microfilarial
load [67], as well as indirect mechanisms such as immunological reactions induced by the par-
asite or the presence of yet-to-discovered pathogens within the parasite microbiome.

In the interim, onchocerciasis elimination efforts must be strengthened in endemic regions
with a high incidence and prevalence of epilepsy. Moreover, as an effort to reduce epilepsy
stigma, communities living in these areas should be explained that the clustering of epilepsy in
certain villages and families is linked to the common exposure to *O. volvulus*-infected black-
flies. Hence, children with NS and other forms of epilepsy should not be isolated.

## Supporting information

**S1 Table. Village matching of epilepsy cases and controls.**
(DOCX)

**S1 Dataset. Dataset with the epilepsy cases and the controls.**
(CSV)

## Acknowledgments

To the study participants, community members and leaders for acceptance to participate.
Health authorities in Morogoro Region and Ulanga District for their cooperation and support
to the study teams. To the data collectors.

## Author Contributions

**Conceptualization:** Luís-Jorge Amaral, Athanas D. Mhina, Robert Colebunders.

**Data curation:** Bruno P. Mmbando.

**Formal analysis:** Luís-Jorge Amaral, Bruno P. Mmbando, Robert Colebunders.

**Funding acquisition:** Robert Colebunders.

**Investigation:** Dan Bhwana, Athanas D. Mhina, Bruno P. Mmbando.

**Methodology:** Bruno P. Mmbando, Robert Colebunders.

**Project administration:** Bruno P. Mmbando.

**Resources:** Robert Colebunders.

**Software:** Bruno P. Mmbando.

**Supervision:** Bruno P. Mmbando, Robert Colebunders.

**Validation:** Dan Bhwana, Bruno P. Mmbando.

**Visualization:** Luís-Jorge Amaral.

**Writing – original draft:** Luís-Jorge Amaral, Robert Colebunders.

**Writing – review & editing:** Luís-Jorge Amaral, Dan Bhwana, Athanas D. Mhina, Bruno P. Mmbando, Robert Colebunders.

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
