## [Decision Letter · Decision Letter 0]

23 Mar 2023

Dear Dr Colebunders,

Thank you very much for submitting your manuscript "Nodding Syndrome, a case-control study in Mahenge, Tanzania: Onchocerca volvulus and not Mansonella perstans as a risk factor" for consideration at PLOS Neglected Tropical Diseases. As with all papers reviewed by the journal, your manuscript was reviewed by members of the editorial board and by several independent reviewers. The reviewers appreciated the attention to an important topic. Based on the reviews, we are likely to accept this manuscript for publication, providing that you modify the manuscript according to the review recommendations. 

Overall, the manuscript is well written, but requires some revisions. Particularly, Methods section requires additional clarifications.

Sincerely,

Angela Monica Ionica, Ph.D.

Academic Editor

Eva Clark

Section Editor

Overall, the manuscript is well written, but requires some revisions. Particularly, Methods section requires additional clarifications.

Reviewer's Responses to Questions

**Key Review Criteria Required for Acceptance?**

**Methods**

-Are the objectives of the study clearly articulated with a clear testable hypothesis stated?

-Is the study design appropriate to address the stated objectives?

-Is the population clearly described and appropriate for the hypothesis being tested?

-Is the sample size sufficient to ensure adequate power to address the hypothesis being tested?

-Were correct statistical analysis used to support conclusions?

-Are there concerns about ethical or regulatory requirements being met?

Reviewer #1: Objectives of the study is clearly articulated with a testable hypothesis. 

The authors report that informed consent was obtained, however detail report is missing. Please specify (preferably under a separate sub-heading) the details of IRB (name and #) that approved the study.

Reviewer #2: Please refer to the general comments below

Reviewer #3: The objectives of this study are clearly stated even if there is a slight problem of semantics. Indeed, the authors sometimes speak of a "causal" role when it is an observational study. They need to change some sentences and be careful about the conclusions drawn. The design of the study is adapted to the objective. The population is well described and the sample size seems correct. However, the authors should indicate how and why this size was chosen.

The statistical analyses are also adequate for the purpose, but the authors should be more specific in their method. In particular, the different models used and the variables included in these models. 

Ethical and regulatory requirements are well respected.

Reviewer #4: -Are the objectives of the study clearly articulated with a clear testable hypothesis stated? YES

-Is the study design appropriate to address the stated objectives? YES

-Is the population clearly described and appropriate for the hypothesis being tested? YES

-Is the sample size sufficient to ensure adequate power to address the hypothesis being tested? YES

-Were correct statistical analysis used to support conclusions? YES

-Are there concerns about ethical or regulatory requirements being met? YES

**Results**

-Does the analysis presented match the analysis plan?

-Are the results clearly and completely presented?

-Are the figures (Tables, Images) of sufficient quality for clarity?

Reviewer #1: No concern

Reviewer #2: Please refer to the general comments below

Reviewer #3: The analyses presented correspond well to the analysis plan in the method. In general, the results are simple and well presented. Some of the results are presented in easy-to-read tables except for Table 3, which does not clearly indicate the models used. An improvement in the presentation of the method would improve the understanding of some results. 

Moreover, no figures are necessary for a good understanding of the results.

Reviewer #4: -Does the analysis presented match the analysis plan? YES

-Are the results clearly and completely presented? YES

-Are the figures (Tables, Images) of sufficient quality for clarity? YES

**Conclusions**

-Are the conclusions supported by the data presented?

-Are the limitations of analysis clearly described?

-Do the authors discuss how these data can be helpful to advance our understanding of the topic under study?

-Is public health relevance addressed?

Reviewer #1: M. perstans infection was assessed by examining dried blood for microfilariae under a microscope. Need to clarify the use of anti-parasitic diethylcarbamazine and albendazole or doxycycline. Prior treatment could affect the presence of M. perstans and lower the risk of active infection. Please discuss if prior treatment of M.perstans was assessed.

Reviewer #2: Please refer to the general comments below

Reviewer #3: The conclusions are in line with the results except for the semantic problem mentioned above. The limitations of the study are well established and relevant. The authors' discussion is well conducted. Although it seems clear that they have a strong view on the issue, they have good arguments and reasoning to try to explain their results. Moreover, this well-conducted discussion is totally focused on public health and will certainly lead to further research and public health actions on this topic.

Reviewer #4: -Are the conclusions supported by the data presented? YES

-Are the limitations of analysis clearly described? YES

-Do the authors discuss how these data can be helpful to advance our understanding of the topic under study? YES

-Is public health relevance addressed? YES

**Editorial and Data Presentation Modifications?**

Reviewer #1: minor revision

Reviewer #2: Please refer to the general comments below

Reviewer #3: (No Response)

Reviewer #4: Abstract 

- Lines 13-14: I suggest you change epilepsy with onchocerciasis-associated epilepsy (OAE) since OAE includes epilepsy, NS and Nakalanga syndrome. Here, your statement seems like epilepsy is synonymous to NS. Your new statement should read “Cases of onchocerciasis-associated epilepsy (OAE) were identified in villages affected by NS in Mahenge, Tanzania, and matched with 14 controls without OAE of the same sex, age and neighbourhood.”

- Line 19-20: Similar to the comment above change epilepsy with OAE. Also, you meant were men and women, respectively instead of were men, respectively; please correct. Your new statement should be “A total of 113 OAE cases and 132 controls were enrolled, of which 56 (49.6%) and 64 (48.5%) were men women, respectively”. 

- Line 20: Please add “respectively” after controls.

- Line 26: Please remove the word “primary” since there is no definitive evidence that OV is the primary casual factor for NS; only association exists. 

- Line 26-27: Please remove the statement “The primary risk factor for NS remains onchocerciasis”. See the previous comment. 

- Lines 29-36: Please change “epilepsy” with “OAE”.

- Line 37: This statement is too strong “In conclusion, onchocerciasis remains the most likely cause of NS”. Please consider to revise. 

Introduction

- Lines 30-40: Please add “s” to 1960. It should be in “1960s”.

- Lines 46-48: You should either describe the complete features of the OAE (epilepsy, NS and Nakalanga syndrome) or just mention that OAE include epilepsy, NS and Nakalanga syndrome. Stunted growth and delayed puberty are not the only features of Nakalanga syndrome that makes it distinct, NS also presents with similar features.

Methods

- Lines 91-92: Please revise the statement “Epilepsy cases and controls without epilepsy were people who screened positive or negative for epilepsy in the previous door-to-door surveys, respectively” to “OAE cases and controls were people who screened positive or negative for OAE in the previous door-to-door surveys, respectively”

- Lines 101-103: Please add “thoracic dysmorphia” and “endocrine dysfunction” to the features of Nakalanga syndrome.

Results

- Please change “epilepsy” with “OAE” throughout where necessary. See previous comments.

Discussion

- Line 183: “1960s”.

- Line 185: change “epilepsy” with “OAE”. Also, for consistency, present cases before controls.

- Please change “epilepsy” with “OAE” throughout where necessary.

- Line 207: Please change “palpitation” with “palpation”.

Conclusion

Lines 286-287: Please change “the most likely” to “a possible”.

**Summary and General Comments**

Reviewer #1: In the Discussion line 194-195: “…Ov16 seropositive persons were also more likely to have a nuclear family member with epilepsy, suggesting that the association is likely related to onchocerciasis exposure.” Furthermore, the authors note that genetics might not play an important role in clustering of epilepsy. Among cases, the percentage of individuals with family history of epilepsy in another immediate family member and nuclear family member is 18.9% and 17.7% respectively, where as among in-Law with epilepsy is 7.1%. Thus, overall incidence of epilepsy is higher among related (1st and 2 nd degree relatives) family members than unrelated. 

Thus, the clustering of epilepsy in family could be due to a genetic predisposition and not due to common parasitic exposure. Another possibility is that there could be increased genetic risk within families for Onchocerca parasite infection. The study did not evaluate this relationship and thus, authors should consider modifying their statement.

In Discussion, line 237-245, the authors suggests reverse causation as a reason for the association between M.perstans and nodding syndrome in South Sudan. Similar argument could be made for increase prevalence of O.volvulus among individuals with NS. In this case-control study, it cannot be ruled out if the association between Onchocerca v. and epilepsy is due to increased risk of infection in persons with epilepsy. One could consider a scenario where persons with epilepsy have an increased functional disabilities and immobility and thus, are more likely to be bitten by vector and at increased risk of OV infection. Request authors to consider such a reverse causation and discuss.

Reviewer #2: Dear Authors,

I have read with interest your manuscript title: Nodding Syndrome, a case-control study in Mahenge, Tanzania: Onchocerca volvulus and not Mansonella perstans as a risk factor.

I appreciate your efforts towards understanding nodding syndrome, a disorder that has devastated poor communities in northern Uganda, and South Sudan in recent times.

I have a few suggestions that may help improve further the quality of the manuscript: I will refer to Onchocerca volvulus as OV, and Mansonella perstans as MP.

1. The title of the study seems to suggest the putative risk factors are “either OV or MP” and nothing else! But it may be that of all risk factors, this study investigated OV and MP and of the two, OV is the risk factor. Be that as it may, the title may not have to suggest the central theme of the findings. You may want to review the title to minimize any form of an “either or” fallacy.

2. In the methods section – both abstract and main text, you refer to having conducted logistic regression analyses, and you report that you adjusted for sex, age and residence, and yet these are also the variables you matched on during recruitment. You may want to explain further why you did not use “conditional logistic regression” but ordinary logistic regression, and why you are adjusting or controlling for the same variables you already match on! Refer to line 137 for example. Second, you may want to explain why you matched on these variables during recruitment, and what was the ratio of the matching? 1:1 or 1:2 etc. and the reasons you chose that ratio for your match.

3. The results may be totally different, the same or similar when a different analytical approaches are used – conditional versus ordinary logistic regressions. I therefore think that further review of the results and discussions would be appropriate after the analytical approaches have been clarified on. But just briefly, you may want to review and report mean with standard deviations and median with interquartile ranges. Note that in line 136 mean is reported with IQR.

Regards

Reviewer #3: COMMENTS

General comment

The authors of the paper "Nodding Syndrome, a case-control study in Mahenge, Tanzania: Onchocerca volvulus and not Mansonella perstans as a risk factor" conducted a study to investigate whether a parasite (Mansonella perstans) previously associated with NS in Sudan was also associated with NS in Tanzania. In view of the data known to date on the subject, all explorations with an etiological aim on NS are of great scientific interest given the current level of our knowledge on this subject. The authors are therefore to be congratulated on the completion of this work

Overall the work is well written and the method seems to have been good even if on reading it some details are missing to improve the understanding of the paper. In addition, some of the conclusions drawn do not always seem to be in line with the work carried out. Please pay attention to this. I will start with general comments on the work and then move on to more specific comments in the manuscript.

From my perspective, there is a small discrepancy between the title (which refers to both Onchocerca volvulus (OV) and Mansonella perstans (M.perstans)) and the objective (which seems to focus on M.perstans). Please modify either one and keep it in mind throughout the work, especially in the conclusions you draw.

Throughout the work, (introduction and conclusion of the abstract; introduction of the work etc.) you say that you were trying to determine whether the M.perstans could play the sole causal role in the development of NS. How could you prove this with this type of epidemiological study? I would suggest that you were looking to see if it could be a factor associated with the occurrence of NS. Indeed, the term "causal" cannot be used in an observational survey.

In the section method you say that the controls were matched to the cases on some criteria but can you tell us more? Was it a one-case, one-control match? If so, why were there 113 cases for 132 controls? How was the number of subjects needed for the Mansonella perstans study calculated?

In general, it would be important to give a little more detail in the method and to be precise. This problem also causes problems in understanding the results.

Below are some minor revisions in the form of specific comments.

Specific comments

Comment 1

In the introduction section, line 47, you state in the text NS for "nodding seizures" and later you use it to refer to "nodding syndrome". Do you consider a syndrome and a seizure to be the same? Or is this a mistake? If so, please correct.

Comment 2

Line 120 (methods): Can you indicate the sensitivity and specificity of the test used for M.perstans at this level? Even if you want to say it again in the discussion.

Comment 3

Can you elaborate on the method? Did you run several models (one model for each variable, adjusted for age, sex and type of village) or were all variables included in the same model? Also, in the results, you give an OR for all epilepsies and an OR for NS only. Is it the same model? The same adjustment variables?

In lines 128 and 129 you mention M.perstans among the variables included in the regression model yet you have no cases of M.perstans! Can you be more precise? In other words, why include it in the models?

When you talk about exposure to onchocerciasis, are you referring to OV16 or the skin manifestations of the disease? Be specific each time.

The answer to these questions may be in Table 3, which assumes that each variable has been adjusted only for age, sex and village. Is this the case? 

If yes, why did you not make a model taking into account all relevant variables? 

If not, what exactly are the variables included in the model(s)?

In any case, please put this information more clearly in the method and if possible present the results of the more complete models.

Comment 4

In discussion: line 195, you explain that the fact that they are neighbours of the cases could explain why a considerable proportion of the in-laws have epilepsy. Of the in-laws with epilepsy in your sample, what percentage live in the neighbourhood of their NS case?

Comment 5

Discussion: Line 246 to 249. You mention poverty as a possible explanation for the link that has been found between M. perstans and NS in Sudan. Is your study population not also poor? And in other places where NS is found? So I don't understand your argument.

Reviewer #4: The study is of importance for the scientific community and the paper is well written. If all comments addressed by the authors, the paper is suitable for publication.

PLOS authors have the option to publish the peer review history of their article (what does this mean?). If published, this will include your full peer review and any attached files.

Reviewer #1: No

Reviewer #2: No

Reviewer #3: No

Reviewer #4: No

Figure Files:

Data Requirements:

Reproducibility:

References

---

## [Decision Letter · Decision Letter 1]

8 May 2023

Dear Dr. Colebunders,

Thank you very much for submitting your manuscript "Nodding Syndrome, a case-control study in Mahenge, Tanzania: Onchocerca volvulus and not Mansonella perstans as a risk factor" for consideration at PLOS Neglected Tropical Diseases. As with all papers reviewed by the journal, your manuscript was reviewed by members of the editorial board and by several independent reviewers. The reviewers appreciated the attention to an important topic. Based on the reviews, we are likely to accept this manuscript for publication, providing that you modify the manuscript according to the review recommendations. 

The comments of the reviewers have been properly addressed, but further clarifications are required.

Sincerely,

Angela Monica Ionica, Ph.D.

Academic Editor

Eva Clark

Section Editor

The comments of the reviewers have been properly addressed, but further clarifications are required.

Reviewer's Responses to Questions

**Key Review Criteria Required for Acceptance?**

**Methods**

-Are the objectives of the study clearly articulated with a clear testable hypothesis stated?

-Is the study design appropriate to address the stated objectives?

-Is the population clearly described and appropriate for the hypothesis being tested?

-Is the sample size sufficient to ensure adequate power to address the hypothesis being tested?

-Were correct statistical analysis used to support conclusions?

-Are there concerns about ethical or regulatory requirements being met?

Reviewer #1: (No Response)

Reviewer #2: Are the objectives of the study clearly articulated with a clear testable hypothesis stated? The objective are clearly stated and testable.

Is the study design appropriate to address the stated objectives? Yes, it is appropriate.

Is the population clearly described and appropriate for the hypothesis being tested?

Yes, the population has been clearly described.

Is the sample size sufficient to ensure adequate power to address the hypothesis being tested?

I have not seen efforts towards stating what the power of the study would be to identify a statistically significant difference between variables being tested by cases - control, status. Some statement of power and sample size calculation will help readers appreciate the study findings better.

I find aspects of matching ratio in the response to reviewers comments but can hardly find these in the main manuscript where they should actually be well embedded as the manuscript is what the readers will interface with once the manuscript is published.

Were correct statistical analysis used to support conclusions? Yes conditional logistic regressions model have now been reported. However, table 3 that seems to have the main issues being reported on, does not quite match what are reported in the text. There is no clear labeling of table 3 with crude and adjusted OR columns. In the text, there seems to be report of both crude or univariate as well as adjusted ORs. Perhaps the table could be made more conventional and easier to read.

Maybe a biostatistician would be required to review the manuscript and ensure the reported ORs are appropriate for what are being discussed.

Are there concerns about ethical or regulatory requirements being met? None.

Reviewer #3: (No Response)

Reviewer #4: The study objectives are clearly articulated, with appropriate study designs addressing the objectives. Study population and sample size are well described and correct statistical analysis were used. Ethical issues were well addressed.

**Results**

-Does the analysis presented match the analysis plan?

-Are the results clearly and completely presented?

-Are the figures (Tables, Images) of sufficient quality for clarity?

Reviewer #1: (No Response)

Reviewer #2: Does the analysis presented match the analysis plan?

Yes

Are the results clearly and completely presented?

I have challenges following the reporting of the results especially where the odds ratios are being reported; mainly in table 3. I have alluded to this earlier on (see above). Not that the results in the abstract in the online system still has the odd results as per ordinary logistics regressions.

Are the figures (Tables, Images) of sufficient quality for clarity? Yes

Reviewer #3: (No Response)

Reviewer #4: The study results are clearly and well presented.

**Conclusions**

-Are the conclusions supported by the data presented?

-Are the limitations of analysis clearly described?

-Do the authors discuss how these data can be helpful to advance our understanding of the topic under study?

-Is public health relevance addressed?

Reviewer #1: (No Response)

Reviewer #2: Are the conclusions supported by the data presented?

For the most part, yes. But what if there were sizeable proportions of controls with evidence of M perstans but no M perstans among the cases - wat would your conclusions have looked like on the main hypothesis that OV is but not M perstans? This is not a counterfactual situation but real. Both cases and controls have 0.0% M perstans? What does that mean should controls have had M perstans and cases did not?

Are the limitations of analysis clearly described?

There maybe need to explicitly report that the study only two measured putative risk for epilepsy and nodding syndrome. Whatever were not measured in the cases and controls may not have balanced out just by virtue of matching on 3 variables. Issues of trauma, and other risk factors out forward by other scholars could not be assessed and this ought to come out clearly as a limitation before the string conclusion that OV is a risk factor. What if something else other than M Perstans lurks behind OV, and that something else not measured to be adjusted for?

Do the authors discuss how these data can be helpful to advance our understanding of the topic under study? Very much so.

Is public health relevance addressed? Yes but some about the use of treated bed nets and stigma seem quite extant.

Reviewer #3: (No Response)

Reviewer #4: Conclusion and recommendation are clearly stated.

**Editorial and Data Presentation Modifications?**

Reviewer #1: (No Response)

Reviewer #2: (No Response)

Reviewer #3: (No Response)

Reviewer #4: (No Response)

**Summary and General Comments**

Reviewer #1: (No Response)

Reviewer #2: (No Response)

Reviewer #3: General comment

We thank the authors for taking into account the comments of the different authors, which has greatly improved the quality of the paper and thus its relevance. The clarification of the methodological aspects brings a better understanding of what has been done.

However, some questions, suggestions and comments remain.

Comment 1

Regarding the association between epilepsy and a family history of epilepsy, it is good that you have mentioned the role of genetics in your discussion. However, your discussion does not explain (or hypothesise) why the association between epilepsy in general is "much" stronger with a second-degree relative than with a first-degree relative. Indeed, whether one considers the geographical proximity of the families or the genetic aspect, one would expect the association to be stronger with first-degree relatives. Similarly, why do you think the presence of epilepsy in in-laws is "much" stronger with NS than with other types of epilepsy? This point was not raised in your discussion either, yet it seems to me to be an important point in your results

Question: Have you considered the possibility that there is a gene (or genetic predisposition) that predisposes to both NS and OV infection? This would explain the epidemiological association found between NS and OV? What do you think?

Comment 2

In Table 3, regarding the "number of years spent in the village", you do not indicate for which "year gap" the OR is calculated. I assume that it is the OR for a one year gap. However, given the medians, I wonder if a one-year difference is meaningful for comparing two people in this population. I suggest you redo the model by comparing two individuals with a one standard deviation years difference in time spent in the village.

Also in Table 3, why did you not run a multivariate conditional regression model on each epilepsy with the following covariates: Ov16, family history of epilepsy and time spent in the village? Is this also a missing data problem? If not, I think it should be done and these results would be more accurate like the association between family history of epilepsy and epilepsy that disappears when the tests (Ov16) are taken into account and that you mentioned in the discussion.

Reviewer #4: (No Response)

PLOS authors have the option to publish the peer review history of their article (what does this mean?). If published, this will include your full peer review and any attached files.

Reviewer #1: No

Reviewer #2: No

Reviewer #3: No

Reviewer #4: No

Figure Files:

Data Requirements:

Reproducibility:

References

---

## [Editor Report · Decision Letter 2]

5 Jun 2023

Dear Dr. Colebunders,

We are pleased to inform you that your manuscript 'Nodding Syndrome, a case-control study in Mahenge, Tanzania: Onchocerca volvulus and not Mansonella perstans as a risk factor' has been provisionally accepted for publication in PLOS Neglected Tropical Diseases.

Best regards,

Angela Monica Ionica, Ph.D.

Academic Editor

Eva Clark

Section Editor

---

## [Editor Report · Acceptance letter]

13 Jun 2023

Dear Dr Colebunders,

We are delighted to inform you that your manuscript, "Nodding Syndrome, a case-control study in Mahenge, Tanzania: Onchocerca volvulus and not Mansonella perstans as a risk factor," has been formally accepted for publication in PLOS Neglected Tropical Diseases.

Best regards,

Shaden Kamhawi

co-Editor-in-Chief

Paul Brindley

co-Editor-in-Chief
